# The Effect of Race/Ethnicity and MED12 Mutation on the Expression of Long Non-Coding RNAs in Uterine Leiomyoma and Myometrium

**DOI:** 10.3390/ijms25021307

**Published:** 2024-01-21

**Authors:** Tsai-Der Chuang, Nhu Ton, Shawn Rysling, Drake Boos, Omid Khorram

**Affiliations:** 1Department of Obstetrics and Gynecology, Harbor-UCLA Medical Center, Torrance, CA 90502, USA; tchuang@lundquist.org; 2The Lundquist Institute for Biomedical Innovation, Torrance, CA 90502, USA; nhu.ton@lundquist.org (N.T.); shawn.rysling@lundquist.org (S.R.); drake.boos@lundquist.org (D.B.); 3Department of Obstetrics and Gynecology, David Geffen School of Medicine at University of California, Los Angeles, CA 90024, USA

**Keywords:** lncRNA, leiomyoma, MED12 mutation, race, miRNA

## Abstract

The objective of this study was to elucidate the expression of long non-coding 
RNA (lncRNA) in leiomyomas (Lyo) and paired myometrium (Myo) and explore the 
impact of race and MED12 mutation. Fold change analysis (Lyo/paired Myo) indicated 
the expression of 63 lncRNAs was significantly altered in the mutated group but 
not in the non-mutated Lyo. Additionally, 65 lncRNAs exhibited an over 1.5-fold 
change in the Black but not the White group. Fifteen differentially expressed 
lncRNAs identified with next-generation sequencing underwent qRT-PCR confirmation. 
Compared with Myo, the expression of *TPTEP1*, *PART1*, *RPS10P7*, *MSC-AS1*, *SNHG12*, *CA3-AS1*, *LINC00337*, *LINC00536*, *LINC01436*, *LINC01449*, 
*LINC02433*, and *LINC02624* was significantly higher, while 
the expression of *ZEB2-AS1*, *LINC00957*, and *LINC01186* was significantly lower. Comparison of normal Myo with diseased Myo 
showed significant differences in the expression of several lncRNAs. Analysis 
based on race and Lyo MED12 mutation status indicated a significantly higher 
expression of *RPS10P7*, *SNHG12*, *LINC01449*, *LINC02433*, and *
LINC02624* in Lyo from Black patients. The expression of *TPTEP1*, 
*PART1*, *RPS10P7*, *MSC-AS1*, *LINC00337*, *LINC00536*, *LINC01436*, *LINC01449*, *LINC02433*, and *LINC02624* was higher, while *LINC01186
* was significantly lower in the MED12-mutated group. These results 
indicate that Lyo are characterized by aberrant lncRNA expression, which is 
further impacted by race and Lyo MED12 mutation status.

## 1. Introduction

Leiomyomas are benign fibrotic tumors characterized by excess accumulation of extracellular matrix (ECM), increased cell proliferation, and inflammation [1,2,3]. Ovarian steroids play a central role in stimulating their growth and progression [4], and Black women have a higher prevalence, size, number of tumors, and symptom severity as compared to White and Hispanic women [5,6,7]. Accumulated genomic studies in leiomyomas have indicated the existence of genetic heterogeneity in these tumors, including chromosomal rearrangements and gene mutation such as *HMGA2* (high-mobility group AT-hook 2), *COL4A5/6* (collagen, type IV, alpha 5 and alpha 6), *MED12* (mediator complex subunit 12), and *FH* (fumarate hydratase), which may be related to leiomyoma development and progression [8,9,10]. Recent studies have focused on the role of MED12 mutation in the pathophysiology of uterine leiomyomas [11,12,13,14]. MED12 serves as a crucial element within the mediator complex, which, in humans, consists of 30 subunits. This complex plays a vital role in the transcription process, acting as a transcription coactivator for RNA polymerase II and participating in the regulation of a diverse set of genes [15,16]. Dysregulation of MED12 is found in many human cancers [16]. Somatic MED12 mutation in exon 2 occurs at a frequency of 60 to 80% in leiomyomas and has a functional role in their initiation and progression, potentially through its influence on many important pathways, including the Wnt/β-catenin signaling, hedgehog signaling, sex steroid receptor signaling, and transforming growth factor (TGF)-β receptor signaling pathways [12,13,14,17,18]. We have previously reported on the differential expression of protein-coding genes, some of which are influenced by race and MED12 mutation of the tumor [19,20]. 

Significant evidence has accumulated indicating a role for lncRNAs in tumorigenesis [21,22], and recent profiling studies using next-generation RNA sequencing have revealed the aberrant expression of non-coding RNA (ncRNA), including lncRNAs, in leiomyomas [23,24,25,26,27,28,29]. LncRNAs are single-strand RNA molecules >200 nucleotides in length and transcribed from different genomic loci representing bidirectional, intergenic, intronic, sense, or antisense regions of protein-coding genes [30,31]. LncRNAs can modulate gene expression through various mechanisms, including functioning as a “molecular sponge” or competing endogenous RNA (ceRNA), and through epigenetic mechanisms such as alterations in chromatin structure, recruitment of epigenetic enzymes, chromatin looping, recruitment of transcription factors or RNA polymerase II, and the release of transcriptional repressors [22,32,33,34,35]. In previous studies, we reported on the differential expression of lncRNAs and sncRNAs, which have 17–20 nucleotides and are further classified as microRNAs (miRNAs, 17–22 nucleotides), small nuclear/nucleolar RNA (snRNA and snoRNA, 70–120 nucleotides), and PIWI-interacting RNA (piRNA, 26–33 nucleotides) [25,36] in a small set of specimens [23,25]. Furthermore, we reported that a special class of long noncoding RNAs, the super-enhancer lncRNAs (SE-lncRNAs), which are transcribed from super-enhancer (SE) genomic loci forming an RNA:DNA:DNA triplex with multiple anchor DNA sites within their sequence, are differentially expressed in Lyo [29]. Functionally, in in vitro studies, we demonstrated the role of lncRNAs MIAT and XIST in Lyo extracellular matrix accumulation through their sponge effects on miR-29c- and miR-200c-, respectively [26,27]. The objective of this study was to elucidate the expression of lncRNA in a large sample set of Lyo and matched Myo specimens and to determine the role of any MED12 mutation of Lyo and race on their expression. 

## 2. Results

### 2.1. Differential Expression of Race/Ethnicity- and MED12 Mutation-Associated Long Non-Coding RNA Transcripts in Leiomyoma and Matched Myometrium

To elucidate the differential expression profile of lncRNAs in Lyo and paired Myo, we performed next-generation RNA sequencing (NGS) using RNA isolated from 19 paired leiomyomas. Following normalization, Hierarchical clustering and TreeView analysis resulted in identification of 4135 lncRNAs whose expression was altered, with increased expression of 1174 lncRNAs and decreased expression of 748 lncRNAs by 1.5-fold or greater in Lyo as compared to paired Myo (Figure 1A). After filtering through the volcano plot analysis, 1068 lncRNAs exhibited upregulation, while 718 lncRNAs displayed downregulation in leiomyomas as compared to their corresponding myometrium (fold change ≥ 1.5 and *p* < 0.05; Figure 1B). Following this analysis, principal component analysis (PCA) was conducted to assess the similarity in RNA transcript patterns between the two groups. The data’s high reliability was confirmed through k-mean analysis, which clustered the samples accordingly (Figure 1C).

We proceeded to investigate the variance in expression of lncRNAs between specimens with MED12 mutations and those without mutations. This analysis, performed as fold change (Leiomyoma/paired Myometrium), resulted in the identification of 3443 lncRNAs with altered expression, of which, the expression of 870 lncRNAs was increased, while the expression of 746 lncRNAs was decreased by 1.5-fold or greater in MED12-mutated specimens as compared with non-mutated specimens. Hierarchical clustering and TreeView analysis effectively categorized these transcripts into distinct groups (Figure 2A). Additionally, we identified 63 lncRNAs that exhibited more than 1.5-fold change (either up or down) specifically in the mutated specimens and with no change in the non-mutated specimens, as depicted in the heatmap (Figure 2B). The Gene Ontology (GO) and KEGG (Kyoto Encyclopedia of Genes and Genomes) pathway enrichment analysis for these 63 uniquely expressed lncRNAs in the mutated group indicated predominant involvement in cancer-associated cell signaling regulation (Figure 2C).

The comparison of the differential expression of lncRNAs based on race resulted in the identification of 3457 lncRNAs with altered expression, of which the expression of 851 lncRNAs was increased, while the expression of 876 lncRNAs was decreased by 1.5-fold or greater in the Black group as compared with the White group. Hierarchical clustering and TreeView analysis effectively categorized these lncRNAs into distinct groups (Figure 3A). We identified 65 lncRNAs that exhibited more than a 1.5-fold change (either up or down) specifically in the Black but not the White group, as illustrated in the heatmap (Figure 3B). The Gene Ontology (GO) and KEGG (Kyoto Encyclopedia of Genes and Genomes) pathway enrichment analysis for these 65 lncRNAs revealed that these lncRNAs were predominantly involved in the regulation of cell cycle and several cell-signaling pathways (Figure 3C).

### 2.2. Validation of Race/Ethnicity- and MED12 Mutation-Associated Long Non-Coding RNA transcripts in Leiomyoma and Matched Myometrium

We next selected 15 novel lncRNAs from the NGS analysis and confirmed their expression by qRT-PCR in 10 non-diseased myometrial samples and in 69 Lyo with their matched Myo from premenopausal patients, including the same tissues that were used for RNAseq. Among these 15 lncRNAs, the expression of *TPTEP1*, *PART1*, *RPS10P7*, *MSC-AS1*, *SNHG12*, *CA3-AS1*, *LINC00337*, *LINC00536*, *LINC01436*, *LINC01449*, *LINC02433* and *LINC02624* was significantly higher, while the expression of *ZEB2-AS1*, *LINC00957* and *LINC01186* was significantly lower in Lyo as compared to matched Myo (Figure 4). When Lyo levels were compared with Myo from non-diseased uteri, the same pattern of change was noted, although the degree of overexpression was more exaggerated for *TPTEP1*, *PART1*, *RPS10P7*, *MSC-AS1*, *SNHG12*, *LINC00536*, *LINC01436*, and *LINC01449* (Figure 4). The comparison of lncRNA levels in Myo from diseased uteri with Myo from non-diseased uteri revealed overexpression of *TPTEP1*, *RPS10P7*, *MSC-AS1*, *SNHG12*, *ZEB2-AS1*, *LINC01436*, and *LINC01449* and reduced expression of *CA3-AS1*, *LINC00337*, *LINC02433*, and *LINC02624* in the diseased Myo (Figure 4). When lncRNA levels were compared as fold change (Leiomyoma/paired Myometrium) in MED12-mutated and non-mutated specimens and Lyo from Black and White patients, the expression of *TPTEP1*, *PART1*, *RPS10P7*, *MSC-AS1*, *LINC00337*, *LINC00536*, *LINC01436*, *LINC01449*, *LINC02433*, and *LINC02624* was found to be significantly higher, while the expression of *LINC01186* was significantly lower in the MED12-mutated group (Figure 5). Moreover, *RPS10P7*, *MSC-AS1*, and *LINC01186* expression were significantly altered in the MED12-mutated group but minimally changed in the non-mutated group (Figure 5). The expression analysis based on race revealed that the expression of *RPS10P7*, *SNHG12*, *LINC01449*, *LINC02433*, and *LINC02624* was significantly higher in Lyo from Black women as compared with white women (Figure 6). Among these lncRNAs, the expression of *RPS10P7* and *SNHG12* in Lyo, as compared with their matched myometrium, was significantly higher in the Black group but minimally altered in the White group (Figure 6). Further comparisons among the three race/ethnicity groups were made based only on expression levels in the Lyo and in the Myo (Figure 7). A comparison of leiomyoma expression of lncRNAs revealed significantly higher levels of *LINC00536* in the Black and Hispanic as compared to the White group and significantly higher levels of *LINC01186* in the Hispanics as compared to the White group (Figure 7). A comparison of myometrial expression of lncRNAs revealed significantly lower levels of *SNHG12* in the Black as compared to the White group and significantly higher *ZEB2-AS1* and *LINC01186* expressed in the Hispanics as compared to the White group (Figure 7). A summary of the analysis from Figure 4, Figure 5, Figure 6 and Figure 7 is shown in Table 1.

Based on their extensive interaction with miRNAs, we selected two lncRNAs, *SNHG12* and *LNC01449*, for functional analysis using the Cytoscape software version 3.10.1 (Figure 8). This analysis was constructed based on ENCORI/starBase (Encyclopedia of RNA Interactomes) [37,38] and demonstrated the highlighted associations between target transcripts which were predominantly involved in the regulation of cell signaling and glycolysis. The analysis showed the involvement of many proteins such as MAP3K7, RPS6KB1, PPP2R2A, RPS6KB2, and HK2 in this interactive network, all of which are well established in leiomyoma pathogenesis [39,40,41,42,43], and other proteins such as ULK1, ELAVL1, SHC3, INSR, and IRS2, which are novel and require further investigation.

## 3. Discussion

Our present study indicates that lncRNAs are aberrantly expressed in Lyo, confirming our earlier study, which was based on a small sample set [23]. We identified a set of lncRNAs which were altered significantly in MED12-mutated Lyo and in Lyo from Black women, but minimally changed in non-mutated Lyo, and in Lyo from White women. We selected 15 lncRNAs (*TPTEP1*, *PART1*, *RPS10P7*, *MSC-AS1*, *SNHG12*, *CA3-AS1*, *LINC00337*, *LINC00536*, *LINC01436*, *LINC01449*, *LINC02433*, *LINC02624*, *ZEB2-AS1*, *LINC00957*, and *LINC01186*) from the NGS analysis and detected their expression with qRT-PCR in 10 non-diseased myometrial and 69 paired leiomyoma specimens. Our results confirmed that the expression of these lncRNAs was consistent with the NGS analysis, and that their expression was affected by the presence of MED12 mutation (*TPTEP1*, *PART1*, *RPS10P7*, *MSC-AS1*, *LINC00337*, *LINC00536*, *LINC01436*, *LINC01449*, *LINC02433*, *LINC02624*, and *LINC01186*) and race (*RPS10P7*, *SNHG12*, *LINC01449*, *LINC02433*, and *LINC02624*).

A strength of this study was that comparisons of lncRNA levels were made both with matched Myo from the diseased uteri and with non-diseased Myo in patients not using hormonal medications prior to surgery. A recent study [44] suggested the use of Myo from non-diseased uteri to compare with Lyo is more advantageous than a comparison made with diseased Myo because of potential effects of leiomyomas on the adjacent Myo. Our data showed that the comparison of gene expression between Lyo and adjacent Myo is similar to the comparison with non-diseased Myo, although for the upregulated lncRNAs, the magnitude of change was more pronounced for most lncRNAs when comparison was made with non-diseased Myo. This finding, along with the significant differences in expression of several lncRNAs in diseased versus non-diseased Myo, supports the notion that the presence of Lyo in and of itself could potentially affect the expression of lncRNA in the adjacent Myo, as previously suggested [44]. However, due to immense heterogeneity in fibroid tissues and our finding demonstrating a similar pattern of differential expression for most lncRNAs using diseased versus non-diseased Myo, the commonly used practice of comparing gene expression levels in Lyo with adjacent Myo is supported.

LncRNAs have been demonstrated to regulate gene expression at various levels through their interaction with miRNAs, mRNAs, DNA, and protein and by acting as epigenetic regulators in stem cell pluripotency and specific lineage commitment [45,46]. In a cell- and tissue-specific fashion, miRNAs and lncRNAs employ these mechanisms to regulate diverse cellular activities under physiological conditions. Additionally, they play a pivotal role in an extensive array of disorders, encompassing tumorigenesis, tumor progression, invasion, and metastasis, as well as cancer diagnosis and prognosis [33,45,46]. The functional role of lncRNAs in leiomyoma pathogenesis has recently been explored [28]. Cao et al. reported on the upregulation of the lncRNA *H19* in leiomyomas and its association with elevated *HMGA2* expression. Because of this linkage, the expression of several genes associated with cell proliferation, inflammation, and extracellular matrix (ECM) deposition was altered [47]. Our recent findings revealed increased expression levels of the lncRNAs *XIST* (X-inactive specific transcript) and *MIAT* (myocardial infarction-associated transcript) in leiomyomas [26,27]. These lncRNAs function as sponges for miR-29c and miR-200c, leading to an upregulation of their respective targets, which include COL3A1, COL1A1, FN1, and TGF-β3 [26,27]. In addition, the expression of *MIAT* correlated with the MED12 mutation status of the leiomyomas, and knock down of MIAT in leiomyoma smooth muscle cells’ spheroids blocked the effects of TGF-β3 on the induction of COL1A1 and COL3A1 expression [27].

Several of the lncRNAs identified in this study have been shown to be involved in tumorigenesis. Among these, TPTEP1 was reported to suppress diabetic retinopathy by reducing oxidative stress [48], inhibiting stemness and radio resistance [49], and it exhibited substantial antitumor effects in several cancers, including hepatocellular carcinoma (HCC) and non-small-cell lung cancer (NSCLC) [50,51]. In addition, recent studies indicated that the expression level of *CA3-AS1* was decreased in gastric cancer and colorectal cancer [52,53]. The overexpression of *CA3-AS1* suppressed the proliferation, invasion, and metastatic capabilities of both gastric cancer and colorectal cancer cells through modulation of the miR-93/BTG3/PTEN axis [52,53]. In contrast to the findings in malignant tumors, leiomyomas which are benign expressed higher levels of *TPTEP1* and *CA3-AS1*. PART1, also known as prostate androgen-regulated transcript 1, was identified in exosomes and affected tumor progression via the miR-17-5p/SOCS6 axis and miR-302a-3p/CDC25A axis [54,55]. Another lncRNA, *MSC-AS1*, promoted cancer progression in gastric, colorectal, hepatocellular, melanoma, and ovarian cancer [56,57,58,59,60,61], and was shown to up-regulate CDK14 expression by acting as a molecular sponge for miR-29b-3p [62]. Our group and others have reported that leiomyomas, as compared with matched myometrium, have decreased expression of the miR-29 family, which plays essential roles in regulating the ECM and cell proliferation through epigenetic mechanisms [63,64,65]. *SNHG12*, also known by other names (*LNC04080*, *ASLNC04080*, *C1orf79*, *LINC00100*, *PNAS-1230*), was overexpressed in several cancers, including non-small-cell lung cancer, renal cell carcinoma, and prostate cancer, and its levels correlated with tumor size, progression, and metastasis [66]. This lncRNA was found to impair DNA damage repair, leading to lesional DNA damage, vascular senescence, and accelerated atherosclerosis [66]. The dysregulation of this lncRNA in Lyo might have relevance to its pathogenesis, as prior studies have shown the aberrant expression of DNA repair enzymes and genomic instability in Lyo [67].

Several lncRNAs *LINC00337*, *LINC00536*, and *LINC01436*, which were overexpressed in Lyo in a MED12 mutation-dependent manner, have been shown to have a role in tumorigenesis in other tissues. *LINC00337* was upregulated in pancreatic ductal adenocarcinoma (PDAC) and facilitated PDAC cell proliferation, along with cell cycle transition, by acting as an E2F1 co-activator through its binding to E2F1 [68]. *LINC00337* acts as an oncogenic lncRNA via its effect on EZH2, to repress p21 and promote gastric cancer cell proliferation [69], and through the miR-1285/YTHDF1 axis, to promote the progression of lung adenocarcinoma [70]. In both colorectal cancer (CRC) and non-small-cell lung cancer (NSCLC), this lncRNA plays a role in promoting tumorigenesis, angiogenesis, and tumor progression. It achieves this by recruiting DNMT1 to suppress the expression of CNN1 and TIMP2 [71,72]. Thus, *LINC00337* may be involved in the stimulation of cell proliferation in leiomyomas, since leiomyomas express higher levels of E2F1, EZH2, and DNMT1 [65,73]. *LINC00536* has been demonstrated to enhance the development and progression of breast cancer through the miR-214/ROCK1 axis and the miR-4282/CENPF axis [74,75]. Additionally, it plays a role in promoting hepatocellular carcinoma via the miR-203b/VEGFA axis and contributes to the progression of bladder cancer by modulating the Wnt3a/β-Catenin signaling pathway [76,77]. Leiomyomas are also characterized by activation of the WNT/β-catenin pathway, which is correlated with MED12 mutation status [13]. *LINC01436* has been shown to be involved in gastric cancer cell proliferation, apoptosis, metastasis, and radio resistance through upregulation of MAPK1 and FBOX11 via sponging and epigenetically silencing miR-585 [78,79]. Thus, LINC01436 may be involved in leiomyoma progression. Our NGS analysis and in vitro confirmation showed the expression of *ZEB2-AS1*, *LINC00957*, and *LINC01186* was significantly lower in Lyo as compared to paired Myo. The lncRNA, *ZEB2-AS1*, is a natural antisense transcript that corresponds to the 5’ UTR of zinc finger E-box binding homeobox 2 (ZEB2). It has been reported to promote tumorigenesis and cancer progression in numerous types of cancer, including gastric, lung, colon, pancreatic, and breast cancer [80]. In an in vivo mouse model of cardiac hypertrophy, the expression levels of *ZEB2-AS1* were elevated, and they altered the progression of cardiac hypertrophy through downregulation of PTEN (phosphatase and tensin homolog) [81]. Our results showed the expression of *LINC01186* was downregulated in leiomyomas. Recent studies also demonstrated a lower level of *LINC01186* in lung cancer and papillary thyroid carcinoma (PTC). This lncRNA was involved in the regulation of EMT and cell proliferation, migration, and invasion in both cancer cells [82,83].

More recently, a role for MED12 in tumor initiation has been proposed, as evidenced by the development of leiomyoma-like tumors in transgenic mice with a uterine conditional mutation in MED12 [8,84]. Leiomyomas with a MED12 mutation are smaller in size and are frequently located subserously [85]. MED12 mutation-negative leiomyomas display copy number alterations in several other mediator complex subunits, such as MED18, MED8, CDK8, and the long non-coding RNA, *RNA340* (*CASC15*). Mutation in the MED12 gene has been shown to disrupt its ability to activate cyclin C-dependent CDK8 [86] and its CDK19 stimulatory activity [87]. Furthermore, the silencing of MED12 in the immortalized human uterine fibroid cell line (HuLM) inhibited Wnt/β-catenin signaling and sex steroid receptor signaling, as well as proteins associated with cell cycle and fibrosis [13]. Our results demonstrated that the expression of several lncRNAs was affected by the presence of a MED12 mutation, which could be contributing to the differences in tumor size and progression in mutated leiomyomas as compared with non-mutated leiomyomas and merits further investigation.

Race plays an important role in leiomyoma severity and progression, with Black patients having larger-size and more numerous leiomyomas, a greater severity of symptoms, and earlier onset [2,88]. Previous research has presented evidence elucidating potential mechanisms contributing to the racial disparity in tumor development and progression, including a higher incidence of vitamin D deficiency in Black patients [89,90]; higher levels of aromatase (CYP19), progesterone receptor A (PR-A), and lower levels of retinoic acid receptor-α (RARA) [89,90]; and increased polymorphism of the genes ER (estrogen receptor), CYP17 (cytochrome P450C17α), and COMT, which are involved in estrogen synthesis, in tumors from Black women [89,90]. Recent reports from our laboratory have revealed a dysregulation of protein-coding and noncoding genes in a race-dependent manner in Lyo, including genes related to tryptophan metabolism [91,92] and SE-lncRNAs/mRNA pairs [29]. In addition, several miRNAs, including miR-200c, miR-21, miR-23b, and Let-7s, were dysregulated significantly in a race-specific manner in leiomyomas [93,94]. In the present study, we showed a race-dependent expression of several lncRNAs, which could be a contributing factor to differences in the expression of protein-coding genes and the racial disparity in fibroid symptomatology.

In summary, our data provide a comprehensive expression profile of lncRNAs in leiomyomas and demonstrate the influence of race/ethnicity and MED12 mutation on their expression, indicating their relevance to fibroid pathogenesis. Our findings also revealed that the differential expression pattern of lncRNAs was similar when comparing lncRNA expression levels in Lyo with adjacent matched Myo versus non-diseased Myo. The aberrant expression of lncRNAs in Lyo through various mechanisms such as interaction with miRNAs, mRNAs, DNA, and proteins could influence the expression of protein-coding genes associated with cell proliferation, ECM accumulation, and inflammation in Lyo.

## 4. Materials and Methods

### 4.1. Myometrium and Leiomyoma Tissues Collection

To reduce the variance among Lyo, tumors between 3 to 5 cm in diameter and with intramural location were obtained from premenopausal women (n = 79; aged 30–54) who underwent hysterectomy for symptomatic Lyo at Harbor-UCLA Medical Center. Normal myometrium was also obtained from surgical cases performed for non-Lyo indications such as prolapse and other urogynecologic disorders. Both gross, histologic and ultrasound studies did not show the presence of Lyo in these 10 specimens. Prior approval from the Institutional Review Board (18CR-31752-01R) at the Lundquist Institute was obtained. Informed consent was obtained from all the patients participating in the study who were not taking any hormonal medications for at least 3 months prior to surgery. The tissues were snap-frozen and stored in liquid nitrogen for further analysis as previously described [25,95]. The racial distribution for the 10 non-diseased myometrial specimens was White (N = 1) and Hispanic (N = 9), and for the 69 pairs was White (n = 8); Black (n = 24); Hispanic (n = 33), and Asian (n = 4).

### 4.2. MED12 Mutation Analysis

Genomic DNA from leiomyomas and paired myometrial specimens from premenopausal women (n = 69) was extracted from 100 mg of freshly frozen tissue using MagaZorb DNA Mini-Prep Kit (Promega, Madison, WI, USA) according to the manufacturer’s protocol. PCR amplification and Sanger sequencing (Laragen Inc. Culver City, CA, USA) was performed to investigate the MED12 exon 2 mutations using the primer sequences in the 5′–3′ direction: sense, GCCCTTTCACCTTGTTCCTT and antisense, TGTCCCTATAAGTCTTCCCAACC, as previously described [27]. The 19 pairs of tissues used for next-generation RNA sequencing were from 11 MED12 mutation-positive and 8 MED12 mutation-negative leiomyomas. The mutation analysis of the specimens (n = 69) indicated that 46 leiomyomas had the MED12 mutations (46/69 pairs; 66.7%) with no mutations in the myometrium. Missense mutations in MED12 exon 2 were the most frequent alteration (38/46 pairs), followed by in-frame insertion–deletion type mutations (8/46 pairs). The missense mutations in exon 2 included c.130G>C (p.Gly44Arg) (5/38 pairs), c.130G>A (p.Gly44Ser) (7/38 pairs), c.130G>T (p.Gly44Cys) (2/38 pairs), c.131G>C (p.Gly44Ala) (3/38 pairs), c.131G>A (p.Gly44Asp) (15/38 pairs), c.131G>T (p.Gly44Val) (5/38 pairs), and c.128A>C (p.Gln43Pro) (1/38 pairs). The MED12 mutation rate was 55.6% (5/9 pairs) in White patients, 82.6% in Black (19/23 pairs), and 60.6% in the Hispanic group (20/33 pairs).

### 4.3. RNA Sequencing and Bioinformatic Analysis

Total RNA was extracted from leiomyoma and matched myometrium (n = 19) using TRIzol (Thermo Fisher Scientific Inc., Waltham, MA, USA). RNA concentration and integrity was determined using a Nanodrop 2000c spectrophotometer (Thermo Scientific, Wilmington, DE, USA) and Agilent 2100 Bioanalyzer (Agilent Technologies, Santa Clara, CA, USA) as previously described [29,96]. Samples with RNA integrity numbers (RIN) greater than or equal to 9 were used for library preparation. One microgram of total RNA from each tissue was used to produce strand-specific cDNA libraries using the Truseq (Illumina, San Diego, CA, USA) according to manufacturer’s instructions. The RNA sequencing was carried out at Novogene Corporation Inc., (Sacramento, CA, USA) and bioinformatics analysis was performed as previously described [19,20]. To visualize the strength of differential gene expression, the Hierarchical clustering and TreeView graph, volcano plot, principal component analysis (PCA) plot, and the Gene Ontology (GO) and KEGG (Kyoto Encyclopedia of Genes and Genomes) Pathway Enrichment Analysis plot were made using Flaski and NcPath [97,98], and the interactive network of lncRNA-miRNA-mRNA was analyzed and made with the ENCORI/starBase (Encyclopedia of RNA Interactomes) and Cytoscape software Version 3.10.1 [37,38,99]. Overall, all the differential gene expressions were acceptable for subsequent statistical analysis. The RNA sequencing data is deposited in Gene Expression Omnibus (GEO) database with accession number (GSE224991).

### 4.4. Quantitative RT-PCR

Briefly, 2 μg of RNA was reverse transcribed using random primers for selected genes according to the manufacturer’s guidelines (Applied Biosystems, Carlsbad, CA, USA). Quantitative RT-PCR was carried out using SYBR gene expression master mix (Applied Biosystems) as previously described [25]. The expression levels of selected genes were quantified using Invitrogen StepOne System, with FBXW2 (F-box and WD repeat domain containing 2) used for normalization [100]. All reactions were run in triplicate, and relative mRNA expression was determined using the comparative cycle threshold method (2^−ΔΔCq^), as recommended by the supplier (Applied Biosystems). Values were expressed as fold change compared to the control group. The primer sequences in the 5′–3′ direction used are listed in Appendix A.

### 4.5. Statistical Analysis

Throughout the text, results are presented as mean ± SEM and analyzed with PRISM software (version 10.1.2, Graph-Pad, San Diego, CA, USA). Dataset normality was determined using the Kolmogorov–Smirnoff test. The data presented in this study was not normally distributed and therefore non-parametric tests were used. Comparisons involving two groups were analyzed using the Mann–Whitney test (Figure 5). Kruskal–Wallis test was used for comparisons involving multiple groups (Figure 4, Figure 6 and Figure 7). Statistical significance was established at *p* < 0.05.

## Figures and Tables

**Figure 1 ijms-25-01307-f001:**
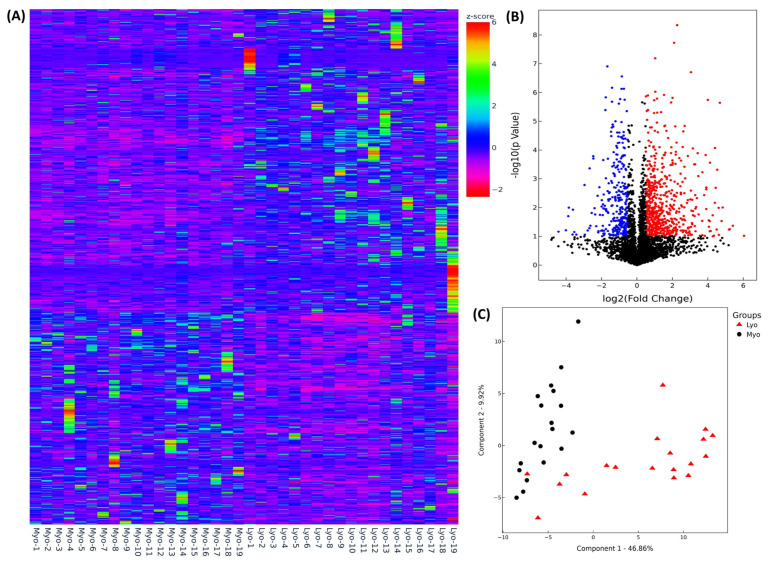
Examination of the heterogeneity and transcriptomic changes of lncRNAs in leiomyomas compared to the myometrium. (**A**) Hierarchical clustered heatmap analysis was performed, illustrating the differentially expressed lncRNAs (fold change ≥ 1.5, *p* < 0.05) in 19 paired leiomyomas and their corresponding myometrium. The color gradient reflects gene expression as z-scores. (**B**) A volcano plot highlights upregulated (red; n = 1068) and downregulated genes (blue; n = 718) with a false discovery rate (FDR) *p*-value < 0.05. (**C**) Principal component analysis (PCA) plot of RNA-seq results from paired leiomyoma and myometrium samples (n = 19). Each dot represents a single sample, with myometrial samples (Myo) shown in black and leiomyoma samples (Lyo) in red.

**Figure 2 ijms-25-01307-f002:**
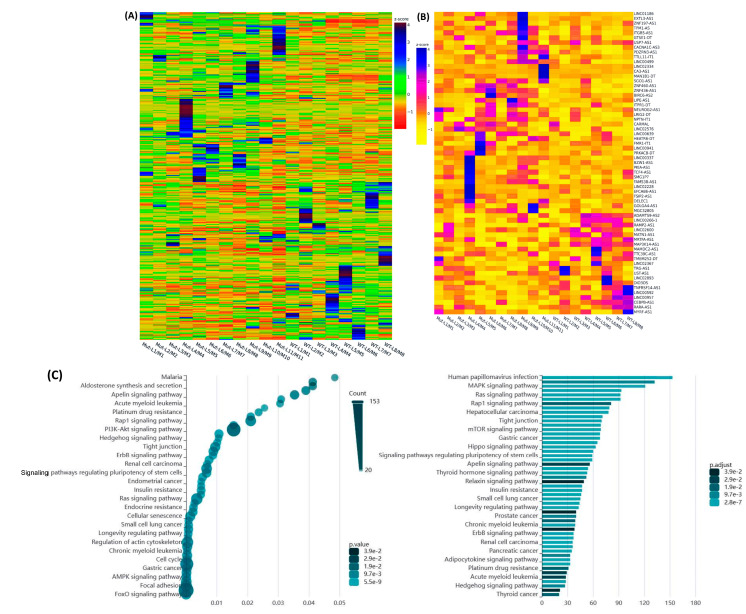
Examination of transcriptomic heterogeneity in lncRNAs within paired specimens based on MED12 mutation status (**A**) was performed utilizing Hierarchical clustered heatmap analysis; fold change (Leiomyoma/paired Myometrium) was assessed for MED12-mutated (n = 11) compared to non-mutated (n = 8) groups (fold change ≥ 1.5, *p* < 0.05). The color gradient reflects gene expression as z-scores. (**B**) Heatmap highlights the 63 enriched genes (Leiomyoma/paired Myometrium) present in the MED12-mutated (n = 11) but absent in the non-mutated (n = 8) groups (fold change ≥ 1.5, *p* < 0.05). The color gradient represents gene expression levels as z-scores. (**C**) Gene Ontology (GO) and KEGG (Kyoto Encyclopedia of Genes and Genomes) pathway enrichment analysis reveals the interactive set of pathways associated with these 63 lncRNAs.

**Figure 3 ijms-25-01307-f003:**
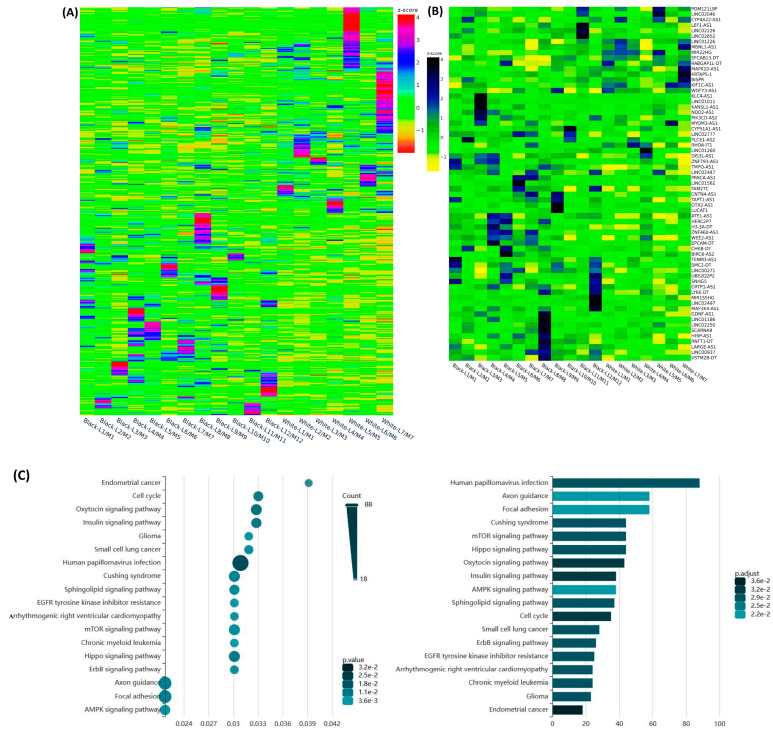
(**A**) Hierarchical clustered heatmap analysis, assessed as fold change (Leiomyoma/paired Myometrium) comparing the Black group (n = 12) with the White group (n = 7) (fold change ≥ 1.5, *p* < 0.05). The color gradient illustrates gene expression as z-scores. (**B**) Heatmap depicts the 65 enriched transcripts (Leiomyoma/paired Myometrium) found in the Black group (n = 12) but not in the White group (n = 7) (fold change ≥ 1.5, *p* < 0.05). The color gradient represents gene expression levels as z-scores. (**C**) Gene Ontology (GO) and KEGG (Kyoto Encyclopedia of Genes and Genomes) pathway enrichment analysis reveals the interactive set of pathways associated with these 65 lncRNAs.

**Figure 4 ijms-25-01307-f004:**
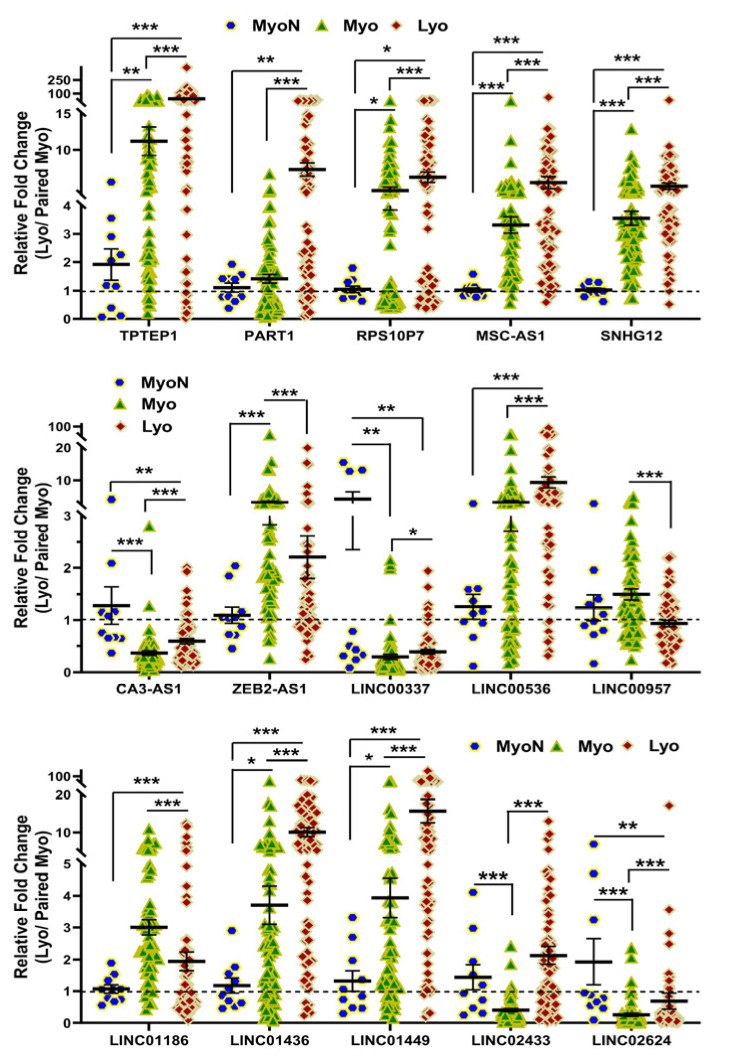
The expression of select lncRNAs in 10 non-diseased myometrial samples (MyoN) and 69 leiomyomas with their paired myometrium by qRT-PCR. The results are presented as mean ± SEM, with *p* values (* *p* < 0.05; ** *p* < 0.01; *** *p* < 0.001) indicated by corresponding lines.

**Figure 5 ijms-25-01307-f005:**
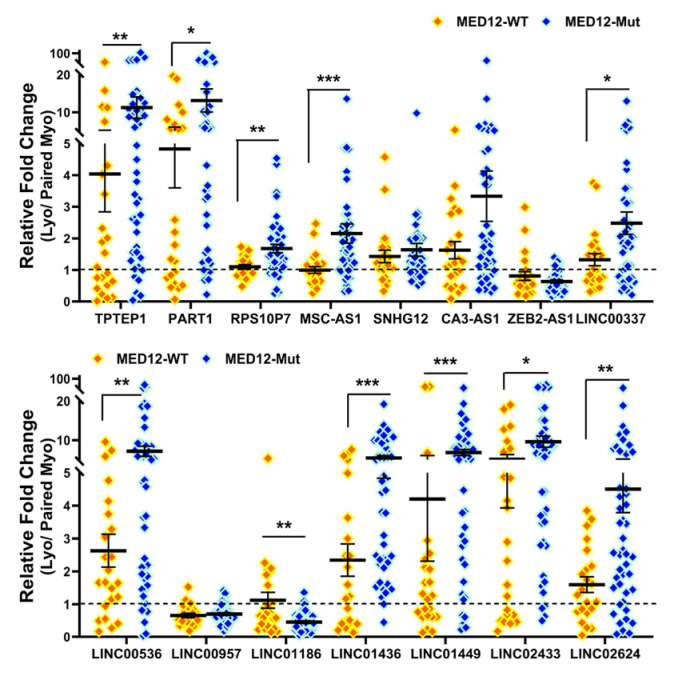
The expression of lncRNAs expressed as fold change (Lyo/paired Myo) in MED12-mutated (n = 46) and non-mutated (n = 23) specimens by qRT-PCR. The results are presented as mean ± SEM, with *p* values (* *p* < 0.05; ** *p* < 0.01; *** *p* < 0.001) as indicated by the corresponding lines.

**Figure 6 ijms-25-01307-f006:**
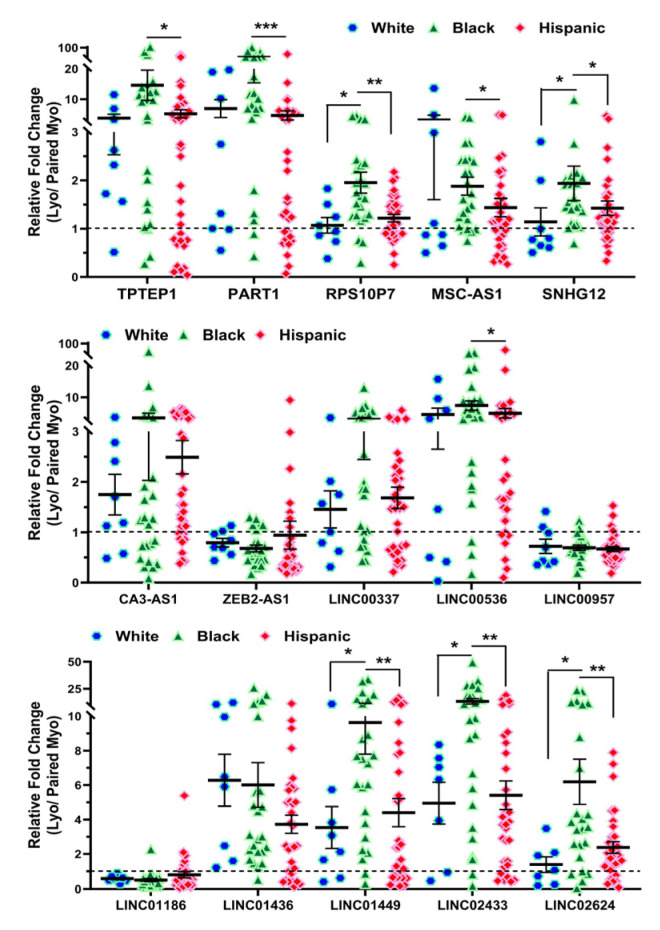
The expression of lncRNAs expressed as fold change (Lyo/paired Myo) in White (n = 8), Black (n = 24), and Hispanic groups (n = 33) by qRT-PCR. The results are presented as mean ± SEM, with *p* values (* *p* < 0.05; ** *p* < 0.01; *** *p* < 0.001) as indicated by the corresponding lines.

**Figure 7 ijms-25-01307-f007:**
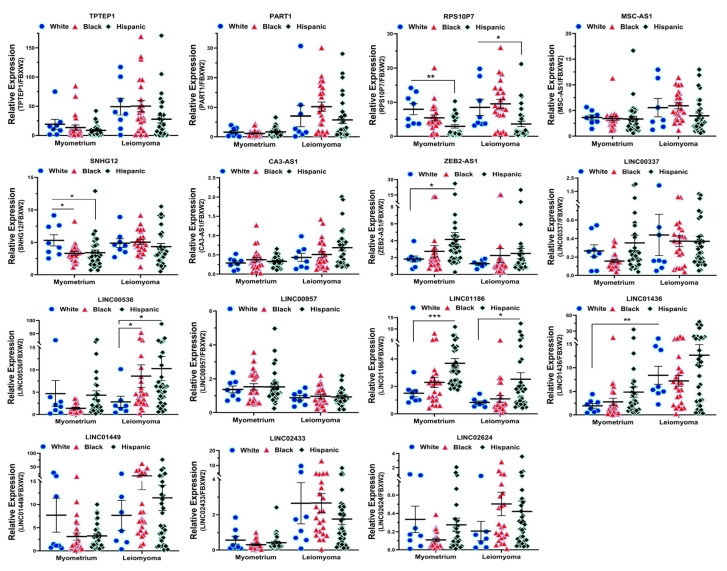
The expression of lncRNAs expressed in the myometrium and leiomyomas in White (n = 8), Black (n = 24), and Hispanic groups (n = 33). The results are presented as mean ± SEM, with *p* values (* *p* < 0.05; ** *p* < 0.01; *** *p* < 0.001) as indicated by the corresponding lines.

**Figure 8 ijms-25-01307-f008:**
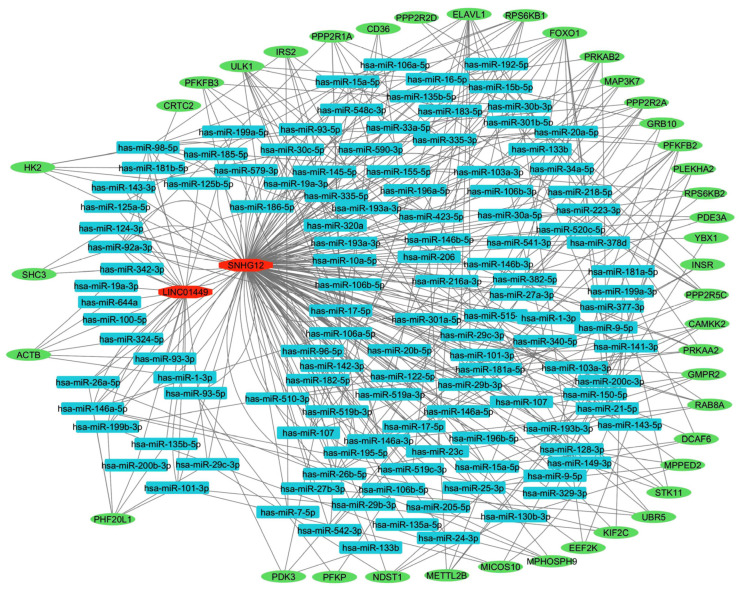
The interactive network (lncRNA-miRNA-mRNA) of lncRNAs *SNHG12* and *LINC01449*. Different colors indicate different RNA molecules (green: mRNAs and blue: miRNAs).

**Table 1 ijms-25-01307-t001:** LncRNAs selected based on the RNAseq analysis according to race and MED12 mutation status.

Symbol	Lyo vs. Myo	MED12-Mut^(Lyo/Myo)^ vs. MED12-WT^(Lyo/Myo)^	Black^(Lyo/Myo)^ vs. White^(Lyo/Myo)^	Black^(Myo)^ vs. White^(Myo)^	Black^(Lyo)^ vs. White^(Lyo)^
TPTEP1	Up (*p* < 0.001)	Up (*p* < 0.01)	No Significance	No Significance	No Significance
PART1	Up (*p* < 0.001)	Up (*p* < 0.05)	No Significance	No Significance	No Significance
RPS10P7	Up (*p* < 0.001)	Up (*p* < 0.01)	Up (*p* < 0.05)	No Significance	No Significance
MSC-AS1	Up (*p* < 0.001)	Up (*p* < 0.001)	No Significance	No Significance	No Significance
SNHG12	Up (*p* < 0.001)	No Significance	Up (*p* < 0.05)	Down (*p* < 0.05)	No Significance
CA3-AS1	Up (*p* < 0.001)	No Significance	No Significance	No Significance	No Significance
ZEB2-AS1	Down (*p* < 0.001)	No Significance	No Significance	No Significance	No Significance
LINC00337	Up (*p* < 0.05)	Up (*p* < 0.05)	No Significance	No Significance	No Significance
LINC00536	Up (*p* < 0.001)	Up (*p* < 0.01)	No Significance	No Significance	Up (*p* < 0.05)
LINC00957	Down (*p* < 0.001)	No Significance	No Significance	No Significance	No Significance
LINC01186	Down (*p* < 0.001)	Down (*p* < 0.01)	No Significance	No Significance	No Significance
LINC01436	Up (*p* < 0.001)	Up (*p* < 0.001)	No Significance	No Significance	No Significance
LINC01449	Up (*p* < 0.001)	Up (*p* < 0.001)	Up (*p* < 0.05)	No Significance	No Significance
LINC02433	Up (*p* < 0.001)	Up (*p* < 0.05)	Up (*p* < 0.05)	No Significance	No Significance
LINC02624	Up (*p* < 0.01)	Up (*p* < 0.01)	Up (*p* < 0.05)	No Significance	No Significance

## Data Availability

Raw data were generated at The Lundquist Institute. Derived data supporting the findings of this study are available from the corresponding author O.K on request.

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
