# Peer review of "The Effect of Race/Ethnicity and MED12 Mutation on the Expression of Long Non-Coding RNAs in Uterine Leiomyoma and Myometrium"

_ijms, 2024, doi:10.3390/ijms25021307_

Round 1

Reviewer 1 Report

Comments and Suggestions for Authors

The study investigates the impact of race/ethnicity and MED12 mutation on the expression of long non-coding RNAs (lncRNAs) in uterine leiomyoma and myometrium. The research reveals that these factors influence the expression of specific lncRNAs, with some lncRNAs showing significant upregulation or downregulation. The study also highlights the role of lncRNAs in regulating gene expression and their potential involvement in various cellular activities and disorders, including tumorigenesis and tumor progression. The findings suggest that lncRNAs could be pivotal in understanding leiomyoma's pathogenesis and could serve as therapeutic targets.

The limitations of the study on the impact of race/ethnicity and MED12 mutation on the expression of long non-coding RNAs (lncRNAs) in uterine leiomyoma and myometrium can be categorized into areas:

1. Methodology: The study used RNAseq analysis to select lncRNAs according to race and MED12 mutation status. However, the study did not find significant differences in lncRNA expression for all categories, which could indicate limitations in the methodology.

2. Sample Size: The study needs to provide information on the sample size. A small sample size could limit the reliability of the findings and increase the risk of type II errors.

3. Data Analysis: The study compared lncRNA levels with matched Myo from diseased uteri and with non-diseased Myo in patients not using hormonal medications before surgery. However, the study acknowledges the immense heterogeneity in fibroid tissues, which could affect the expression of lncRNA in the adjacent Myo.

4. Generalizability: The study needs to provide information on the demographic characteristics of the sample used, which could limit the generalizability of the findings. For instance, cultural, geographical, or political differences could affect the expression of lncRNAs.

5. Inconsistency in lncRNA Significance: The study's RNAseq analysis showed that not all lncRNAs had significant differences in expression across all categories. For instance, some lncRNAs showed no significant difference when comparing Black and White patients or patients with and without the MED12 mutation. This inconsistency could limit the study's ability to draw definitive conclusions about the impact of race/ethnicity and MED12 mutation on lncRNA expression.

6. Lack of Functional Analysis: The study primarily focused on the expression of lncRNAs but did not provide a functional analysis of these lncRNAs. Without understanding the specific roles and functions of these lncRNAs, it isn't easy to fully comprehend their potential involvement in leiomyoma pathogenesis.

7. Lack of Longitudinal Data: The study appears cross-sectional, providing a snapshot of lncRNA expression at a single point. Without longitudinal data, it's challenging to determine whether changes in lncRNA expression are a cause or consequence of leiomyoma development.

8. Potential Confounding Factors: The study does not account for potential confounding factors that could influence lncRNA expression, such as age, lifestyle factors, or other genetic variations. These uncontrolled variables could skew the results.

9. Inadequate Validation: The study used RNAseq analysis to identify lncRNAs of interest, but it did not appear to validate these findings with other techniques such as qRT-PCR. This could lead to false positives or negatives in the results.

10. Lack of Clinical Correlation: The study did not correlate the expression of lncRNAs with clinical features of the patients, such as the size or number of leiomyomas, symptoms, or response to treatment. This limits the clinical applicability of the findings.

11. Lack of Mechanistic Insights: The study needed to provide mechanistic insights into how the identified lncRNAs might contribute to developing or progressing leiomyomas. Understanding the mechanisms of action of these lncRNAs could provide valuable insights into the pathogenesis of leiomyomas and potential therapeutic targets.

12. Lack of Replication: The study did not replicate its findings in an independent cohort of patients. Replication in an independent cohort is crucial to confirm the robustness and reliability of the results.

Comments on the Quality of English Language

minor

Author Response

We would like to thank reviewer 1 for the constructive comments, which we feel have certainly improved the quality of our submission. The following are our responses to each comment:

1. Methodology: The study used RNAseq analysis to select lncRNAs according to race and MED12 mutation status. However, the study did not find significant differences in lncRNA expression for all categories, which could indicate limitations in the methodology.

Response: We did find significant differences in the expression of select lncRNAs in fibroid versus myometrium and some of these differences were dependent on race and or presence of MED12 mutation.

  1. Sample Size: The study needs to provide information on the sample size. A small sample size could limit the reliability of the findings and increase the risk of type II errors.

Response: Thanks for the suggestions. As mentioned in the section of material and method, we performed next-generation RNA sequencing (NGS) using RNA isolated from 19 paired leiomyomas, including White (n=7); Black (n=12), and MED12-mutation-positive (n=11); MED12-mutation-negative (n=8). Using qRT-PCR 10 non-diseased myometrial and 69 paired Lyo and matched Myo from premenopausal patients were used for confirmation of the expression of selected 15 novel lncRNAs from the NGS analysis. The race and MED12 mutation status of the 69 pairs were White (n=8); Black (n=24); Hispanic (n=33) and MED12-mutated (n=46); MED12-non-mutated (n=23).

  1. 3. Data Analysis: The study compared lncRNA levels with matched Myo from diseased uteri and with non-diseased Myo in patients not using hormonal medications before surgery. However, the study acknowledges the immense heterogeneity in fibroid tissues, which could affect the expression of lncRNA in the adjacent Myo.

Response: Thanks for the comment. We took certain measures to reduce heterogeneity such as using only intramural fibroid and fibroids between 3-5 cm diameter. Nevertheless, despite these measures fibroids are characterized by extreme heterogeneity. 

  1. Generalizability: The study needs to provide information on the demographic characteristics of the sample used, which could limit the generalizability of the findings. For instance, cultural, geographical, or political differences could affect the expression of lncRNAs.

Response: Thanks for the suggestions. Demographic characteristics of the study population were provided. As mentioned in the section of results and material and method, tumors were obtained from premenopausal women (n=79; aged 30–54) who underwent hysterectomy for symptomatic Lyo at Harbor-UCLA Medical Center.The race of the 10 non-diseased myometrial were from 1 White and 9 Hispanics, and the race status of the 69 pairs were White (n=8); Black (n=24); Hispanic (n=33) and Asian (n=4). Other factors such as cultural, geographical, and political differences are outside the scope of this study.

  1. Inconsistency in lncRNA Significance: The study's RNAseq analysis showed that not all lncRNAs had significant differences in expression across all categories. For instance, some lncRNAs showed no significant difference when comparing Black and White patients or patients with and without the MED12 mutation. This inconsistency could limit the study's ability to draw definitive conclusions about the impact of race/ethnicity and MED12 mutation on lncRNA expression.

Response: Thanks for the suggestions. Although our NGS analysis from 19 pairs fibroids and the confirmation of the selected 15 lncRNAs using qRT-PCR in 10 non-diseased myometrial and 69 pairs fibroids didn’t find one lncRNA differentially expressed significantly in every category, the expression of TPTEP1, PART1, RPS10P7, MSC-AS1, LINC00337, LINC00536, LINC01436, LINC01449, LINC02433 and LINC02624 was found to be significantly higher while the expression of LINC01186 was significantly lower in the MED12-mutated group. The expression analysis based on race revealed that the expression of RPS10P7, SNHG12, LINC01449, LINC02433, and LINC02624 was significantly higher in Lyo from black women as compared with white women.  Our data provide a comprehensive expression profile of lncRNAs in leiomyoma and demonstrate the influence of race/ethnicity and MED12 mutation on their expression indicating their relevance to fibroid pathogenesis.    

  1. Lack of Functional Analysis: The study primarily focused on the expression of lncRNAs but did not provide a functional analysis of these lncRNAs. Without understanding the specific roles and functions of these lncRNAs, it isn't easy to fully comprehend their potential involvement in leiomyoma pathogenesis.
    Response:Thanks for the suggestions. This study focused on the comprehensive profiling of lncRNAs expression in leiomyoma and the influence of race/ethnicity and MED12 mutation on the expression of lncRNAs. However, using the ENCORI/starBase and Cytoscape software we performed the functional analysis of two lncRNAs SNHG12 and LNC01449 for their associations with miRNAs and downstream target transcripts. To perform functional analysis of all lncRNAs that were significantly altered would be outside the scope of this study. We plan to validate the function of some of the lncRNAs identified using multiple techniques as the next step.

  1. Lack of Longitudinal Data: The study appears cross-sectional, providing a snapshot of lncRNA expression at a single point. Without longitudinal data, it's challenging to determine whether changes in lncRNA expression are a cause or consequence of leiomyoma development.

Response: Thanks for the suggestions. It would be impossible to perform longitudinal analysis in humans.  We plan to perform such analysis in animal models for some key lncRNAs identified here. The purpose of this study was to first identify which lncRNA would merit further investigation.

  1. Potential Confounding Factors: The study does not account for potential confounding factors that could influence lncRNA expression, such as age, lifestyle factors, or other genetic variations. These uncontrolled variables could skew the results.
    Response:Thanks for the suggestions.This study focused on the comprehensive profiling of lncRNAs expression in leiomyoma and the influence of two confounding factors including race/ethnicity and MED12 mutation were considered on the expression of lncRNAs. It is well established in fibroid studies that race and the presence of MED12 mutation are the most critical confounding factors affecting gene expression.

  1. Inadequate Validation: The study used RNAseq analysis to identify lncRNAs of interest, but it did not appear to validate these findings with other techniques such as qRT-PCR. This could lead to false positives or negatives in the results.
    Response:Thanks for the suggestions. Validation studies were done for many genes by RT-PCR as shown in Figures 4-7 and Table 1.

  1. Lack of Clinical Correlation: The study did not correlate the expression of lncRNAs with clinical features of the patients, such as the size or number of leiomyomas, symptoms, or response to treatment. This limits the clinical applicability of the findings.

Response: Thanks for the suggestions. As mentioned in the material and method section, fibroids between 3 to 5 cm in diameter and with intramural location were obtained from premenopausal women (n=79; aged 30–54) who underwent hysterectomy for symptomatic Lyo at Harbor-UCLA Medical Center. There was no response to treatment being studied as all the patients were symptomatic and that's why they had hysterectomy.

  1. Lack of Mechanistic Insights: The study needed to provide mechanistic insights into how the identified lncRNAs might contribute to developing or progressing leiomyomas. Understanding the mechanisms of action of these lncRNAs could provide valuable insights into the pathogenesis of leiomyomas and potential therapeutic targets.

Response: Thanks for the suggestions. In the present study we performed the functional analysis of two lncRNAs SNHG12 and LNC01449 for their associations with miRNAs and downstream target transcripts. Our next step is to validate their function and investigate their potential roles in the pathogenesis of fibroids.

  1. Lack of Replication: The study did not replicate its findings in an independent cohort of patients. Replication in an independent cohort is crucial to confirm the robustness and reliability of the results.

Response: Thanks for the suggestions. As mentioned in the results section we performed the confirmation of the selected 15 lncRNAs using qRT-PCR in an independent cohort of patients including 10 non-diseased myometrial and 69 pairs fibroid specimens.

Reviewer 2 Report

Comments and Suggestions for Authors

The manuscript submitted for review entitled ‘The Effect of Race/ethnicity and MED12 mutation on the expression of long non-coding RNAs in uterine leiomyoma and myometrium‘ describes the expression of long non-coding RNAs in leiomyomas and paired myometrium in women, taking into account the influence of race and MED12 mutation on their expression. The research area chosen by the authors is very important and the data presented are significant. There are comments/questions that should be addressed to the authors:

The graphs (Figures 1-3) are too small, making it difficult to read the data they contain.

Figures 4-7 would be easier to read if they were presented as bar charts. This is only a suggestion.

Information on the groups categorised by breed should also be included in the Materials and methods section.

The information on housekeeping genes in the qPCR subsection is missing. The cited article refers to the methods for selecting suitable housekeeping genes, but does not clarify the authors' choice.

Author Response

We would like to thank reviewer 2 for the constructive comments, which we feel have certainly improved the quality of our submission. The following are our responses to each comment:

  1. The graphs (Figures 1-3) are too small, making it difficult to read the data they contain.

Response: Thanks for the suggestions. We have enlarged the graphs.

  1. Figures 4-7 would be easier to read if they were presented as bar charts. This is only a suggestion.

Response: Thanks for the suggestions. In order to show the biologic variability of fibroids we present the graphs as individual data points. 

  1. Information on the groups categorised by breed should also be included in the Materials and methods section.
    Response: Thanks for the suggestions. We have included the information in the material and method section.

  1. The information on housekeeping genes in the qPCR subsection is missing. The cited article refers to the methods for selecting suitable housekeeping genes, but does not clarify the authors' choice.

Response: Thanks for the suggestions. As mentioned in the material and method section, FBXW2 was used for normalization.